# Tinkering *Cis* Motifs Jigsaw Puzzle Led to Root-Specific Drought-Inducible Novel Synthetic Promoters

**DOI:** 10.3390/ijms21041357

**Published:** 2020-02-18

**Authors:** Aysha Jameel, Muhammad Noman, Weican Liu, Naveed Ahmad, Fawei Wang, Xiaowei Li, Haiyan Li

**Affiliations:** College of Life Sciences, Engineering Research Center of the Chinese Ministry of Education for Bioreactor and Pharmaceutical Development, Jilin Agricultural University, 2888 Xincheng Street, Changchun 130118, China; Aysha_jameel@hotmail.com (A.J.); mohmmdnoman@gmail.com (M.N.); liuweican602@163.com (W.L.); naveed_jla@gmail.com (N.A.);

**Keywords:** synthetic promoter, drought stress, GUS reporter gene, soybean hairy roots, transgenic Arabidopsis

## Abstract

Following an in-depth transcriptomics-based approach, we first screened out and analyzed (in silico) *cis* motifs in a group of 63 drought-inducible genes (in soybean). Six novel synthetic promoters (SynP14-SynP19) were designed by concatenating 11 *cis* motifs, *ABF, ABRE, ABRE-Like, CBF, E2F-VARIANT, G-box, GCC-Box, MYB1, MYB4, RAV1-A*, and *RAV1-B* (in multiple copies and various combination) with a minimal 35s core promoter and a 222 bp synthetic intron sequence. In order to validate their drought-inducibility and root-specificity, the designed synthetic assemblies were transformed in soybean hairy roots to drive GUS gene using pCAMBIA3301. Through GUS histochemical assay (after a 72 h 6% PEG6000 treatment), we noticed higher glucuronidase activity in transgenic hairy roots harboring SynP15, SynP16, and SynP18. Further screening through GUS fluorometric assay flaunted SynP16 as the most appropriate combination of efficient drought-responsive *cis* motifs. Afterwards, we stably transformed SynP15, SynP16, and SynP18 in Arabidopsis and carried out GUS staining as well as fluorometric assays of the transgenic plants treated with simulated drought stress. Consistently, SynP16 retained higher transcriptional activity in Arabidopsis roots in response to drought. Thus the root-specific drought-inducible synthetic promoters designed using stimulus-specific *cis* motifs in a definite fashion could be exploited in developing drought tolerance in soybean and other crops as well. Moreover, the rationale of design extends our knowledge of trial-and-error based *cis* engineering to construct synthetic promoters for transcriptional upgradation against other stresses.

## 1. Introduction

The expansion of world population in combination with an increased risk of natural disasters due to global warming have jeopardized food security [1]. To fulfill the zero hunger challenge, demand for crops like soybean (*Glycine max* (L.) Merrill), a primary source of the world’s supply of vegetable oil and protein as well as feed and pharmaceuticals, has increased. Unfortunately, osmotic stresses, water, and temperature fluctuations and other factors also affect soybean crop against which it lacks tolerance [2]. Plants respond to drought stress through a number of morphological, biochemical, and physiological measures. A reduction in the size of leaf, extension of stem, and proliferation of root distorts plant water link while water use efficiency is declined. The establishment of a ramified root system leading to increased root-to-shoot ratio is one of the strategies to enhance water uptake for photosynthesis under less severe drought [3,4]. Similarly, the phytohormone, ABA (abscisic acid) has a strong link with drought as its de novo synthesis is greatly elevated when the root cells perceive soil water deficit [5]. ABA acting as an intercellular messenger is transported to leaves where it regulates stomatal closure through the guard cells thereby slowing down photosynthesis and other growth-related metabolic activities [6,7,8]. In addition, it also triggers several drought-responsive genes functioning in drought tolerance including those involved in the synthesis of osmoprotectants [9]. Through osmotic adjustment, plants accumulate compounds including betaine, glycine, proline, fructan, inositol, mannitol, and inorganic ions to decrease osmotic potential and ameliorate intracellular water retention under drought stress [10,11,12,13]. These osmolytes also protect plasma membranes and enzymes which face damage by the reactive oxygen species generated when the dynamic equilibrium of ROS production and scavenging is broken [14,15]. Along with osmolytes mentioned, a massive literature also reported the involvement of polyamines (PA)—especially putrescine, spermine, and spermidine—in positive regulation of drought stress. Either endogenous biosynthesis or exogenous application of PA under drought stress improved osmotic adjustment and triggered tolerance-related genes [16].

The current global strategy is to reverse climate change in parallel with exploiting biotechnology and genetic engineering to develop crop varieties which would cope with adverse environmental conditions [17,18]. Transgenesis holds great promise in improving crops at DNA level. An important approach to manipulate crops is their transcriptional modification/upgradation through synthetic promoters. This technology has aided in developing promoters that are optimized to facilitate tightly controlled transgene expression, thus enabling efficient genetic transformation [17,19]. Taking benefit of the modern biotech inventory, we set out to design root-specific drought-inducible promoters according to the protocols we have reported previously [20].

Promoter is the *cis* acting nucleotide sequence located upstream of a gene. To date, numerous promoters have been characterized from plants and viruses and used in the transgenic plants production [21]. Subsequently, promoter studies are fundamental for refining our understanding of gene regulation at transcriptional level and to apply in transgenic crop production. On the basis of transcriptional activity, they are categorized as constitutive, inducible, and spatiotemporal promoters [22]. The well described CaMV 35S (CaMV—cauliflower mosaic virus) promoter which is about 54 bp in length, has been used widely for the constitutive gene expression [23]. In monocots and dicots, CaMV 35S has proved an efficient promoter for transcription initiation, however the constitutive expression is undesirable at times [24]. Numerous *cis* motifs along with the core promoter have been tested in plants [25,26]. As a result, the synthetic sequence is comprised of several specific *cis* elements in an absolutely new orientation but still can initiate transcription [27].

This technology can be used to modify discrete sequences derived from different promoters to generate a unique transcriptional module that contains the functionalities of both the donor and acceptor promoters. The resultant chimeric or recombinant promoter has an enriched genetic constitution that is far superior to its parent promoters, thus delivering greater efficiency using a relatively rapid approach [28,29]. In contrast to the constitutive expression like that of 35S promoters, we at times need stimulus-inducible and root-specific expression of a gene, thus these promoters are specific to root and are induced only under drought stress. A temporarily regulated gene which is expressed in a specific tissue or even in a specific organelle might produce a better effect than a rampant overproduced protein disturbing the environment or clogging the protein synthesis machinery. Synthetic promoters can help control the time, place, and extent of the target gene expression inside an organism. It is becoming more and more important to construct synthetic promoters with differences in tissue specificity, inducibility and strength for different reasons such as: (1) Desired gene expression levels required in plant metabolic engineering research demand the synthetic promoter development with controlled levels of expression. (2) Multiple genes simultaneous co-expression raises a specific trait and in this approach of gene stacking, each gene is expressed by a discrete promoter and thus homology-dependent genes silencing (HDGS) and unwanted recombination are prevented [30]. (3) Plant biotechnology encompasses gene transformation experiments for developing resistance in plants to biotic and abiotic stresses and thus synthetic promoters for respective environmental conditions are indispensable [31].

These advantages have led to increased use of chimeric promoters for targeted gene expression over the past years [32]. Most recombinant promoters aim to achieve maximal transcription by optimizing the copy number and spacing of the key *cis* elements in their sequences [17,33]. Several techniques have been employed to design and test these promoters [17,28,29,33,34]. Approximately 200 such promoters have been successfully developed by engineering different abiotic [35,36] and biotic [17,37] stress-responsive *cis* elements or domains using these techniques. Further information relating to synthetic promoters that facilitate chemical-, light-, and hormone-responsive gene expression or constitutive/tissue-specific gene expression, and the respective *cis* sequences used to design them, was provided in our previous report [20]. The minimal promoter has sequences for binding to the transcriptional factors including the TATA box. The core promoter contains different regulatory sequences like *GA elements* or a coreless sequence along with some core elements like *CA element*, *Inr element*, *CCAAT element*, *Y patch*, and others [38]. The position of the core promoter elements (50 bp away from the transcription start site) relative to the transcriptional start site along the length of a gene is conserved in plants [39].

The spacing and copy number of motifs can be optimized in designing synthetic promoter once the desired motifs are identified, however this needs experimental verification. As the transcription factors attach to their corresponding *cis* elements in an ordered fashion, the *cis* motifs in a synthetic promoter should be properly spaced to get a perfect synergistic effect [40]. The efficiency of the synthetic promoter can be affected by the *cis* motif dosage like in tobacco, a synthetic promoter with double ACGT motifs spaced with 5 nucleotides was inducible by salicylic acid while the same motifs when distanced with 25 nucleotides made it abscisic acid inducible promoter.

We set out to test the hypothesis that *cis* motifs in the promoters of drought-responsive genes in combination with root base could be used in designing root-specific drought-inducible synthetic promoter. We aimed at driving β-*glucuronidase* gene through the novel assemblies in response to drought in soybean transgenic hairy roots as well as Arabidopsis roots. The designed synthetic modules were transformed in soybean hairy roots and Arabidopsis where their tissue-specific inducibility against drought stress was tested. The GUS analysis of both transgenic systems demonstrated that *cis* motifs, their copy number and spacing were appropriate to construct root-specific drought-inducible promoters and the strategy could be reproduced for other type of promoters in crops besides soybean.

## 2. Results

### 2.1. Transcriptome Analyses Determined Drought-Inducible Cis Motifs

Prior to designing synthetic promoter against drought stress, we had to select appropriate *cis* motifs. *Cis* motif families were identified among drought-responsive soybean genes revealed through transcriptome analysis previously [41,42]. Initially, differentially expressed genes (DEGs) were identified in the early and late morning periods (ZT0 and ZT4). The genes that were differentially expressed in each time period or in more than one period during drought stress were selected, making up a list of total 63 genes (Appendix A). Based on hierarchical clustering analysis, these 63 genes made up four groups (A, B, C, and D), as shown in Figure 1. The heat map revealed that majority of soybean genes in cluster 1 and 2 showed markedly high transcript abundance profiles against drought stress as compared to those in cluster 3 and 4. By using the reference gene annotation information for the identification of *cis* motifs, we selected 1000 bp sequence upstream of transcription initiation site as promoter regions of all 63 genes. Using BEST, all the promoter sequences were analyzed to search for drought-inducible conserved sequence motifs. To further refine the output of motif discovery programs, Bio Optimizer [43] was integrated with MEME, AlignACE, CONSENSUS, and BioProspector in BEST [44]. Subsequently, the identified motifs were classified with STAMP [45] which on the basis of similarity, determined mutual relationship of 63 *cis* regulatory sequences to known annotated motifs in AGRIS databases [46,47,48,49,50,51]. To illustrate the similarities among 73 sequence motifs, a tree was generated using MEGA as shown in Figure 2. For easier analysis of motif groups and tree, a cutoff value of 0.035 was set between two nodes, which was the least selected length of a branch used in MEGA. With this cutoff value, 12 motif family binding profiles were obtained. Among the 12 motif groups, 2 groups (XI to XII) showed less significant similarities (higher E-value) to a wide array of *cis* elements. Many *cis* motifs were not really being associated with abiotic stress response. After analysis with BEST software tools—MEME, CONSENSUS, and Bioprospecter—*cis* motifs expressed more than two times were proceeded for further experimental analysis. We found that some motifs were highly homologous to known motifs of the database with minimum E-Value. The height and number of letters per bit represent the degree of conservation of the position. The 73 motifs detected with BEST were further analyzed using STAMP, which not only determined relationship among the motifs but also identified similarities with known motifs. Among 73 motifs clusters, 11 *cis* motifs were selected which are reported to be involved in drought stress. These selected motifs were further analyzed with the PlantPatho database, the results showed that these drought-inducible motifs were active not only in roots but in leaves, shoots and other parts of the plants at different times. Detailed STAMP results can be found in Appendix A.

### 2.2. Motifs Selection and Design of Synthetic Modules

After careful analysis of *cis* motifs from the promoters of dehydration-triggered upregulated genes, the second step was to construct an assembly of synthetic promoter. The 11 *cis* elements (Table 1) [52,53] were selected to design novel synthetic promoters. MEME analyses suggested that *ABRE*, *MYB*, and *E2F-VARIENT* play key role in transcriptional activation and these could form components of synthetic promoters to drive transgene expression [54].

Taking pCAMBIA3301 as a base carrier, we designed six vector constructs for drought-specific expression in soybean hairy roots. Synthetic promoter constructs were prepared by fusing the *cis* motifs upstream of 35S minimal promoter named (SynP14, SynP16, SynP18) while other three constructs (SynP15, SynP17, SynP18) were designed by fusing the same motifs at both sides (3′ and 5′ end) of 35S minimal promoter. The type, order and copy number of all *cis* motifs used in synthetic promoter design are shown in Figure 3. The recombinant constructs were created by substituting the CaMV 35S promoter in pCAMBIA3301 with the designed synthetic promoters as shown in Figure 4. Similarly, the drought inducible rd29A promoter was also inserted by replacing the CaMV 35S between *NcoI* and *HindIII*, which was used as control for drought specific expression.

### 2.3. Agrobacterium-Mediated Transformation of Soybean Hairy Roots and GUS Analyses

Subsequently, after constructing the recombinant plasmids with synthetic modules (SynP14-SynP19) and controls (35S and rd29A), they were transformed into *Agrobacterium rhizogenes* (strain K599) for induction of transgenic hairy roots as described [62]. Prior to agro-infection, K599 transformants were verified through PCR using construct-specific primers (Figure 5) and the PCR-positive cells were cultured.

For regeneration of hairy roots, we infected 5-day old cotyledons (*G. max* cv *Jiyu 72*) seedlings with K599 transformants harboring the constructs as mentioned [63,64]. Two weeks after infection, hairy root had regenerated and were confirmed (whether transgenic) through PCR using synthetic promoter-specific primers. When the transgenic hairy roots were long and strong enough to support the plants, the primary roots were removed. The chimeric plants were rooted in fresh vermiculite and allowed to grow until hairy roots were ~10 cm long. One set of transgenic hairy roots were subjected to GUS histochemical analysis before drought treatment, while another set of transgenic hairy roots was put forward to GUS staining after PEG treatment of chimeric plants. Transgenic hairy roots were excised from ~30 days old chimeric plants for histochemical GUS assay in order to check the inducibility of synthetic promoters against drought. Without drought treatment, no staining was found for any synthetic promoter except 35S control (Figure 6A).

For drought-inducible expression, the transgenic hairy roots of six synthetic constructs were treated with 6% PEG6000 for 72 h. Interestingly, all the synthetic promoters displayed drought-inducible GUS expression, however, SynP14, SynP17, and SynP19 showed partial GUS staining as compared to the other three (SynP15, SynP16, and SynP18) as shown in Figure 6B. The staining results particularly confirmed that some *cis* motifs in these synthetic modules were highly active upstream of the 35S promoter region. We selected the SynP15, SynP16, and SynP18 which displayed higher GUS expression after GUS staining, for fluorometric assay. The 35S was used as control for constitutive expression and rd29A was used as control for drought-inducible expression respectively. For each binary vector construct, three independent hairy roots were further analyzed for transformation efficiency each time. Every independent hairy root was considered as an independent transformation event [65]. There was no GUS expression in roots without the osmotic stress confirming that the promoter elements were only active under drought stimulus thus were drought-inducible.

Whole chimeric soybean plants were also analyzed against drought stress. Transformed plants with SynP16 had significant tolerance against the drought stress. However, chimeric plants with SynP18 hairy roots showed less sustainability as compared to SynP16 but higher than SynP15. So it was further demonstrated that the *cis* motifs used in the SynP16 responded with high efficiency against drought stress as compared to other two promoters. All of the root-specific drought-inducible promoters showed GUS staining however, SynP15, SynP16, and SynP18 were proceeded for subsequent experiments as they exhibited higher level of GUS staining as shown in Figure 6B. For further validation of promoter efficiency, we extracted the total protein from SynP15, SynP16, and SynP18 hairy roots for GUS fluorometric assay using Hitachi 4600 spectrophotometer. The quantification results showed that SynP16 had higher GUS activity as compared to the SynP15 and SynP18 as shown in Figure 6C. In addition, the copy number as well as spacing of those particular *cis* motifs must also have contributed to the better performance.

### 2.4. Functional Validation of Synthetic Promoters in Arabidopsis Thaliana

In order to further validate root-specific drought-inducible expression of synthetic promoters, we successfully transformed three synthetic promoters (which displayed high GUS expression in soybean hairy roots) i.e., SynP15, SynP16, and SynP18 in *Agrobacterium tumefaciens* EHA105 strain for transgenesis of Arabidopsis. After EHA105 transformants were verified through promoter-specific PCR, they were used to stably transform Arabidopsis. Thus T3 (Transgenic 3) generation of Arabidopsis was obtained after double selection (1% basta and promoter-specific PCR) of transgenic Arabidopsis as shown in figure. We tested the two stably transformed homozygous lines for drought inducible GUS expression of synthetic promoters. Twenty days old transgenic *Arabidopsis* were treated with 20% PEG6000 for 24 h to check the efficiency of synthetic modules. Wild type *Arabidopsis* plants (WT) were used as control for this experiment. The whole plant including roots and shoots were subjected to GUS histochemical assay which indicated that SynP15, SynP16, and SynP18 drove significantly high GUS expression in roots of Arabidopsis. Arabidopsis GUS staining assay further confirmed the root-specific drought-inducible expression of synthetic promoters as shown in Figure 7A.

For GUS quantification assay, total protein from roots of Arabidopsis plants subjected to 20% PEG6000 was extracted and analyzed with Hitachi 4600 spectrophotometer. In consistence with hairy roots results, SynP16 in Arabidopsis showed the highest GUS expression as compared to SynP15 and SynP18 shown graphically in Figure 7B. For tissue-specific drought-inducible expression, mature Arabidopsis plants were stressed by withholding water for 10 days and the absence of histochemical staining revealed that the promoters were root-specific. From soybean hairy roots and Arabidopsis GUS assays, it is clear that our designed synthetic modules had root-specific drought-inducible expression.

## 3. Discussion

Modern sequencing technologies in combination with efficient plant transformation methods responded to numerous questions which were previously unknown for scientific community [66,67,68]. Statistics show that 57% of the total oilseed production depends on soybean (*Glycine max* (L.) worldwide. As a consequence of global warming, drought is causing massive losses in many important crops including soybean. Many conventional methods including the genomic and bioinformatics tools have been exploited to understand the mechanism of drought tolerance for the production and selection of drought resistant varieties. Fortunately, current transgenic technology is an efficient tool for soybean crop improvement due to its long breeding cycle and intricate genetics. Soybean transformation approaches like *Agrobacterium*-mediated and biolistics are well-recognized through various published experimental reports [69,70,71,72,73]. Transcriptional upgradation of a crop through synthetic promoter is a biotechnological approach with significant advantages. In promoters, specific *cis* regulatory elements within the promoter region are responsible for tissue-specificity, duration, timing, and strength of gene expression. Various promoters from virus, bacteria, and plants have been modeled to study transgene regulation. We used bioinformatics tools for the detection of novel *cis* regulatory elements that are conserved among the stress-inducible genes. The main goal of this work was the identification of *cis* motifs in drought-inducible genes and exploit in designing synthetic promoters. 

Initially synthetic promoters were designed by the combination of defined *cis* regulatory motifs with a strong constitutive promoter [17,74,75,76] or in combination with a strong promoter by replication of the upstream enhancer domains [77]. Shen et al. (1996) did fundamental work on the development of abiotic stress-inducible synthetic promoters. A 22 bp synthetic promoter ABRC3, was designed containing *ACGT box* (in the same orientation as in the native promoter) and coupling element 3 (CE3) merged with a minimal promoter (Amy64). A single copy showed sufficient response to abscisic acid ABA (approximately nine fold), whereas constructs containing two (44 bp) and three copies (66 bp) showed approximately 20-fold and 25-fold GUS activity compared to the 22 bp promoter following exposure to ABA. Multiple synthetic promoters were designed by interchanging spacing, copy number, and orientation of coupling elements (*CE1/CE3*) and *ACGT* motif [78]. In our study, we used the same strategy to design the synthetic modules with multiple motifs in soybeans by using multiple drought inducible *cis* motifs which would enhance drought-inducible expression specifically in roots.

Kim et al. (2002) designed synthetic promoters to study the effect of four copies of the C/DRE *cis* acting element that was derived from the cor15a gene to check the expression of GUS (β-glucuronidase) under cold and drought stress and found that light signaling is involved in conferring resistance to cold and drought stress in transgenic plants [55]. We combined various *cis* motifs in multiple copies that were conserved among different species against drought stress to get drought-inducible expression in soybean as well as in *Arabidopsis*. Tissue-specific promoters have an advantage for driving the expression of transgenes over constitutive promoters in a specific cell type because they leave the other tissue unaffected. In the last two decades, various tissue-specific promoters have been discovered and characterized in plants. For transgene regulation, fine tuning of *cis* regulatory elements is essential for targeted and accurate functions of promoters. Ishige et al. (1999) reported that in both dicots and monocots, the *G box* motif is responsible in a non-tissue specific manner for high level constitutive expression [79]. However in our study we used multiple copies of *G box* elements in combination with root-specific motifs specifically designed for their drought-inducible expression in SynP16 and SynP18. Our studies proved that combination of drought-inducible motifs with root motif showed precise expression against drought stress in tissue-specific manner.

The promoters currently used for the development of transgenic soybean transformation are limited in number and only few provide tissue-specific expression against multiple stimuli (such as drought, cold, and salt stress). We designed and tested six synthetic promoters for root-specific drought-inducible expression and validated in heterologous systems. Transgenic crops development could be greatly enabled by the ability to computationally design synthetic promoters. To date, all commercial transgenic crops use strong constitutive promoters, which have limited value for multi-trait transgenics for which multiple unique promoters are needed. In addition, it would be of great value to be able to design very short (~100 bases) and strong synthetic promoters for precise transgene expression [80]. In our studies, we designed synthetic promoter with short length of minimum 300 bp to 800 bp that can provide precise drought-inducible expression. Synthetic promoters provide an efficient platform for testing motif functions [17,33]. 

*Cis* motif is a necessary part of synthetic promoters. Among numerous species, defense signaling across species is well conserved. The local gene expression in plants is regulated by several *cis* motifs (*boxes W1, Gst1, W2, S, GCC, D*, and *JERE*) recognized by their specific transcription factors (Dofs, WRKYs, bHLHs, bZIPs, ERFs, MYBs) against pathogen attack in plants [17]. The distinct synthetic promoters with four copies of *cis* motifs were designed and their expression analysis was carried out during multiple pathogen interaction like non-host, compatible, and non-compatible interactions. It is difficult to predict the effect of spacing between single *cis* motif for transgene expression and can only be studied experimentally [81]. A second study about pathogen defense promoter was conducted by the improvement in many parameters like copy number of single motif in promoter and it changed the inducibility and strength of the promoter [17].

We used the same strategy with some modification by addition of root motif to design root-specific promoter. We selected multiple elements obtained after promoter analysis of the upregulated genes during drought stress and combined multiple elements that were found in more than two groups. The copy number and spacing among *cis* motifs are very important for the precise control of gene expression. Four copies of root-specific motif were placed in every synthetic promoter upstream of 35S. The other *cis* motifs that are stress responsive were placed in two copies or one copy each in the synthetic promoters as mentioned. Six recombinant vectors were constructed individually to check the root specific drought inducibility of synthetic promoters. Our study gives the unique pattern of synthetic modules with multiple drought-inducible elements experimentally validated for root-specific expression. The GUS assays of transgenic plants confirmed root-specific expression of synthetic promoters and proved the feasibility of synthesizing tissue-specific promoters in soybean and in Arabidopsis. Meanwhile, these novel synthetic promoters can decrease the defects of original regulatory sequences in tissue-specific manner. They also showed valuable expression in transient expression system of hairy roots and Arabidopsis roots thus can meet the requirements of various applications. 

Introns, particularly the first intron in the 5′ untranslated region (5′UTR), via intron mediated enhancement (IME) can significantly increase gene expression. From literature review, we selected the intron sequence from soybean *elongation factor 1A* (*eEF1A*) gene (GmScreamM8) that was reported for high activity in native promoter [82]. The sequence of synthetic intron part 3 (4xEF4-M8CIN3) 222 bp showed higher expression than other parts. Tetramer elements located upstream of GmScreamM8 core promoter, expressed very high activity for stable expression in soybean hairy roots and transient expression analysis in lima bean cotyledons. The native leader intron included, is active for an interaction between promoter elements and intronic sequences. In our studies, we used the synthetic intron sequence 3 that was reported to be the most suitable for expression analysis in soybean. It was put upstream of the 35S minimal promoter after combination of the *cis* regulatory elements for designing of synthetic promoter.

There are still numerous combinatorial mechanisms of regulatory context and signaling that are largely unknown, which prevent the optimal design of tissue-specific synthetic promoters [33]. Advances in bioinformatics and in-depth studies of plant TF networks and *cis* and *trans* synergistic interactions, could greatly accelerate design strategies for the construction of effective synthetic promoters. A high-throughput promoter designing strategy adhering to all the critical factors mentioned above, combined with in silico methods would be a solution to generate synthetic promoters with tunable transgene expression. The synthesis of tissue-specific promoter in soybean hairy roots proved that method used for the analysis of root-specific synthetic promoter is reliable and can be exploited for the functional analysis of *cis* motifs in the roots for tissue specific expression. The GUS-specific expression in transgenic hairy roots of soybean and Arabidopsis roots showed the reliability of our method. Our study represents a major contribution to the understanding of synthetic promoters, and develops a toolbox of synthetic promoters and promoter elements for fundamental and applied research.

In this study, we demonstrated that out of the six synthetic promoters, SynP16 could be used for root-specific drought-inducible expression in soybean. However, as we have driven only *glucuronidase* gene with these synthetic promoters, we suggest to test the expression of other drought-responsive genes with SynP16. In addition, SynP16 should also be transformed and tested in other important crops (maize, wheat, and rice) to develop drought tolerance in them. Moreover, this study provides tools and approaches for researchers in the field of synthetic biology dealing with crop improvement. As the title of the research paper refers synthetic promoter a ‘jigsaw puzzle’, taking *cis* motifs and other accessory elements as pieces of this puzzle. Nature has intelligently put these pieces in a manner they work best. The *cis* motifs are discrete and each motif can function when put in appropriate position and combination. Thus shuffling these motifs and testing their performance though a trial and error-based method constitute the only way to solve the puzzle in getting the best combination of appropriate *cis* motifs (an ideal synthetic promoter). Furthermore, researchers could easily pick *cis* motifs (Appendix A) of their choice and put them according to our design or their own for constructing more novel synthetic promoters which would be more portable/tunable.

## 4. Materials and Methods 

### 4.1. Selection of Cis Motifs and Designing of Synthetic Modules

In order to obtain optimal drought-responsive genes, the intersection of genes upregulated in both studies (ZT0, Z4 [41]; and R2, V6 [42]) was finally used as the final candidate drought corresponding genes. By processing the chip data in the literature [42] to obtain the drought-responsive gene expression profiles under different conditions, and then clustering the expression profiles, we obtained four clusters. Statistically significant *cis* motifs were identified by an in silico analysis of these 63 drought-responsive genes. A total of 11 drought-inducible motifs were detected which appeared with highest frequency among 73 motif clusters. The selected motifs were common for the majority of analyzed genes. Their significance was evaluated by searching for their presence in transcription factor binding site databases (TRANSFAC, PlantCARE [83], and PLACE [51]). The PathoPlant database was used for in silico analysis of 11 motifs in drought-treated roots ( available online: http://www.pathoplant.de/expresion_analysis.php). This database is used for analysis of *cis* element responsive to various biotic and abiotic stimuli by correlation of sequence occurrences in *A. thaliana* promoters with microarray expression data. Designed synthetic promoter sequences were analyzed using SoftBerry’s Plant Promoter Database (PPD) [84]. Our study needed a very precise site for the transcription start site (TSS) that can be identified from SoftBerry PPD (promoter prediction database). As a result, SoftBerry database is main source for the in silico analysis of designed promoter sequences. We chose the *cis* motifs in multiple numbers that are active in the promoter region of drought resistant genes for the designing of synthetic modules to activate the GUS gene. 

Six vector constructs were designed specifically for GUS expression in transient expression system of soybean transgenic hairy roots. Synthetic modules named SynP14, SynP15, Syn P16, SynP17, SynP18, and SynP19 were constructed by fusing the different sets of multiple *cis* motifs and a root motif with intron sequence from soybean *elongation factor 1A* (*EF1A*) gene (third intron 222 bp) [82] upstream of 35S minimal promoter. For synthetic module SynP14, multiple copies of *cis* motifs and synthetic intron sequence were fused upstream of 35S minimal promoter sequences and in SynP15, the same elements were fused at both the 5′ and 3′ ends of minimal promoter. Similarly, in SynP16, drought inducible synthetic modules were adjusted upstream of 35S promoter and SynP17 contained same module at both sides of 35S minimal promoter region. The third pair of the synthetic module Syn P18 contained another set of *cis* motifs active against drought stress upstream of 35S minimal promoter and SynP19 contained the same motifs at both 3′ and 5′ end of 35S promoter region. However the synthetic intron sequence was placed at the end of synthetic modules upstream of 35S minimal promoter region. 

The designed synthetic modules were fused with 35S minimal promoter by using pCAMBIA 3301 as back bone. Promoters were cloned in pCAMBIA3301 by replacing the CaMV 35S promoter so as to drive *GUS* reporter gene. The newly constructed recombinant plasmids were named SynP14, SynP15, SynP16, SynP17, SynP18, and SynP19. Drought inducible rd29A promoter in pCAMBIA3301 by replacing the 35S promoter named (rd29A) was used as control for drought-inducible expression. For constitutive expression, CaMV 35S promoter in pCAMBIA3301 (35S) was used.

### 4.2. Agrobacterium Rhizogenes-Mediated Transformation of Soybean Hairy Roots

The plasmid DNA was isolated from the transformed cells by using the E.Z.N.A^®^ Plasmid Mini Kit I (Omega Bio-Tek, Doraville, CA). The transformed *Escherichia coli* cells were grown in 10 mL LB media containing the antibiotic kanamycin (100 mg/l) and incubated at 37 °C at 200 rpm for 17 h. The culture was centrifuged in microcentrifuge tube (2 mL) at 13,000 rpm for 10 min and the pellet formed was suspended in 250 µL of solution I and gently vortexed. The freshly prepared 250 µL solution II was added to the 0.2 mL tube, and mixed thoroughly by inverting 4–5 times. The tube was placed at room temperature for 5 min. The sample turned viscous.

To precipitate, 350 µL of solution III was added and mixed by inverting the tube several times. The tube was left for 5 min at room temperature and then centrifuged at 12,000 rpm for 10 min. The supernatant was transferred to a fresh spin column and collection tube and the pellet was discarded. The supernatant was passed through spin column and then add 700 µL volume of HBC wash buffer. The tube was left for 2 min at room temperature and centrifuged at 12,000 rpm for 1 min. The liquid passed in to collection tube was discarded and then extracted plasmid was washed with DNA wash buffer two times. The pellet was air dried and finally the 30 µL of elution buffer (Pre heated 65 °C) was added into tube.

Seeds of soybean (*Jiyu 72* cultivar) were surface sterilized for 13 h with chlorine gas. A single layer of seeds were placed in petri dishes under the fume hood in a vacuum desiccator. The beaker containing sodium hypochlorite solution placed in the desiccator to which HCl was added, and instantly covered with lid and sealed with parafilm. Seeds were kept in the chlorine gas atmosphere for 12–16 h. Then seeds were sown in open pots contained clean vermiculite in growth chamber (16 h light/ 8 h dark, 25 °C, and 50% humidity). We used the vent controlled and clear covered tray (30 cm, 40 cm, and 15 cm high) that is commercially available and used as the ‘humid chamber’ for plants. Sterilized soybean seeds were put at a depth of 1–2 cm into wet vermiculite.

*Agrobacterium* strain(s) that contained the desired promoter constructs were prepared by streaking bacteria from the glycerol stock on to surface of YEP solid media plates contained antibiotics kanamycin 50 mg/l and streptomycin 50 mg/l and incubated at 28 °C for 2 days. The single colony was picked and restreaked on the surface of the fresh plate. The fresh bacterial culture was taken from the plate and suspended in 1 mL of YEP liquid media that contained 15% vol/vol glycerol stock. The suspension of the 200 µL of the bacterial culture spread on the surface of YEP media containing appropriate antibiotics. These plates were incubated overnight at 28 °C and used to infect the five days old soybean seedlings grown on the fresh vermiculite. Unfolded green and healthy cotyledons were used for *Agrobacterium* hairy root transformation. Hypocotyls proximal region or cotyledonary node was stabbed with bacterial paste and/or bacterial suspension. The central part of the hypocotyls was infected carefully with bacterial paste and pushed it inside wounding site. 

Cucumopine type *Agrobacterium rhizogenes* strain K599 [62] was used for induction of transgenic hairy roots to further check the promoter activity. The 35S (Constitutive expression) and rd29A (drought-inducible expression) were used as the positive and negative control respectively. In this experiment, expression vector pCAMBIA3301::GUS (35s) was used to check synthetic promoter function to drive GUS gene expression in hairy root. For all experiments, vermiculite was used for seed germination and for the growth of hairy roots. Infected plants were supplemented with Hoagland nutrient solution at hairy root growing stage.

The infected plants were transferred in to wet vermiculite and the tray was covered with the sterile transparent lid sealed it with the sticky tape to maintain the humidity for formation and growth of the hairy roots. The infected seedlings were watered with half strength Hoagland’s solution once every week. After 2 weeks, hairy roots 1–2 cm in length were emerged from infection site. Plants that contained hairy roots were transferred in to new trays contained fresh vermiculite for root elongation and the lid was open to decrease the humidity. The plants that grow the hairy roots were kept in a growth chamber (humid chamber only) at 12 h light/ 12 h dark at 28/25 °C for 2 weeks. At this time period the hairy roots were long enough to support the plant growth. During this period, water the plants with Hoagland solution. When the length of hairy roots was approximately 5–10 cm long, the primary roots were cut by removing the hypocotyls near wounding site where hairy roots were grown.

### 4.3. PEG Stress Treatment of Chimeric Soybean Plants

Chimeric soybean plants harboring transgenic hairy roots were treated with 6% PEG6000 to drought stress for 72 h. The 35s was used as the positive control for constitutive GUS expression while rd29a as control for drought-specific expression. For each binary vector, three independent in vivo transformation experiments with the same treatments and growth conditions were carried out, and 200 seedlings were transformed and at least 100 hairy roots were analyzed for transformation efficiency for each time. Each hairy root was considered as an independent transformation event.

### 4.4. GUS Histochemical Staining and Fluorometric Analysis of Transgenic Soybean

Histochemical localization of GUS activity was performed with 5-bromo-4-chloro-3-indolyl glucuronide (Xgluc) as a substrate. Soybean transgenic hairy roots were incubated in 0.1 M sodium phosphate buffer (pH 7.0) with 1 mM X-gluc overnight at 37 °C. After the GUS reaction, roots were washed with 70% ethanol prior to observation. The percentage of stained hairy roots was used as an estimation of the efficiency of the transformation method.

Bradford method was used for estimation of total protein concentration in the plant extracts [85]. 0.5 g of transgenic root tissue was ground in GUS extraction buffer for each construct, vortexed and centrifuged at 13000 rpm for 15 min. The supernatant was collected in a fresh 1.5 mL Eppendorf tube and 50 µL of the extract was used to estimate the total protein. Furthermore, protein extract was added to 4 mL of Bradford’s reagent, incubated in dark for few minutes and the total protein was estimated using spectrophotometer at 595 nm. Bovine serum albumin (BSA) 250 µg was used as a standard.

GUS activity of transgenic hairy roots of soybean were determined according to Jefferson (1987) [86]. A 0.5 g tissue was ground in GUS extraction buffer (50 mM sodium phosphate pH 7.0, 10 mM EDTA, 0.1% v/v Triton X-100, 0.1% Sarkosal, and 10 mM 2-mercaptoethanol). The supernatant of crude extracts was incubated in 1 mM 4-methylumbelliferyl-b-D-glucuronide, followed by analysis with spectrophotometer. The 50 µL of the protein extract was added to 0.5 mL aliquots containing GUS assay buffer pre warmed at 37 °C and mixed thoroughly. It was mixed thoroughly. The 20 µL of this mixture was transferred to a microcentrifuge tube containing 0.18 mL stop buffer after regular time intervals (0, 0.5, and 2 h) at room temperature. Fluorometer readings were recorded with excitation at 365 nm and emission at 455 nm using Hitachi 4600 spectrophotometer. The GUS activity was defined as fluorescence units per microgram of soluble protein per hour.

### 4.5. Agrobacterium Tumefacians-Mediated Transformation of Arabidopsis

The newly constructed plasmids (50 ng) of synthetic modules were added to centrifuge tube containing 50 µL of competent cells (EHA105) and the contents were mixed gently. The tube was stored on ice for 30 min and later transferred to a water bath pre heated to 42 °C for 90 s. Then cells were transferred immediately to ice bath for 1–2 min. 900 µL of YEP broth was added to the tube and incubated at 37 °C with shaking at 150–200 rpm in an incubator shaker for 2–3 h. Finally, the Eppendorf tube was centrifuged at 6,000 rpm and pellet was resuspended in 100 µL of YEP broth and plated over YEP solid medium containing 100 mg/l kanamycin. The plates were incubated at 37 °C for 12–16 h. The colonies appeared in the plates were picked in random and the transformants were confirmed using colony PCR with promoter specific primers. Specific primers were used for confirmation of PCR positive colonies.

Wild type *Arabidopsis* (ecotype Columbia) seeds were kindly provided by the Engineering Research Center of the Bioreactor and Pharmaceutical Development, Ministry of Education of Jilin Agricultural University. Arabidopsis plants were grown in pro mix soilless medium mixed with vermiculite in 2:1 ratio at 18 h photoperiod at 21 °C. Seeded pots were placed in a tray covered with a plastic transparent lid (to maintain humidity for freshly sown seeds) and placed at 4 °C for 2 days (for stratification) in order to allow uniform germination. The lid was slightly opened at the seedling stage to allow aeration for growing seedlings and then finally removed after a three-day hardening off period. The plants were watered on alternate days and provided with Miracle-Gro (Scotts company, OH, USA) fertilizer twice a week.

### 4.6. Agrobacterium Tumefaciens Strain EHA105 was Used for Stable Transformation in Arabidopsis Thaliana

Introduction of recombinant vectors (SynP15, SynP16, and SynP18) into EHA105 was performed by the freeze–thaw method [87]. Arabidopsis plants intended for transformation were grown as above except with fewer seeds (4–5) per pot. The plants were grown until the primary bolts appeared. Primary bolts were clipped to allow the propagation of many secondary bolts. After 4–6 days of clipping, plants were transformed using floral dip method.

*Agrobacterium* harboring desired plasmids were grown at 28 ℃ overnight in 7 mL YEP medium supplemented with kanamycin (50 µg/ mL). The following day, the 7 mL culture was used to inoculate 700 mL of YEP medium with appropriate antibiotic and grown overnight. Cultures were started from a 1:100 dilution of overnight cultures and grown approximately for 18–24 h. Next day, the 700 mL culture was centrifuged at 5000 rpm for 10 min and the bacterial pellet was suspended in 5% (w/v) sucrose solution to achieve an OD_600_ prior to use. For floral dip inoculation medium MS media contained 5.0% sucrose and 0.05% (i.e., 500 µL 1-1) Silwet L-77 (OSi Specialties, Inc., Danbury, CT, USA). The inoculum was dispensed into a beaker, plant tips were inserted into suspension to submerge all the tissues above the ground, and plants were then detached from the media after 5 min of mild agitation. Plants dipped in to beaker were removed and put in to tray that was covered with clear plastic to maintain humidity. Plants were kept in dark or in low light overnight and then placed in to normal conditions but kept away from the direct sun light. Lid was removed after 24 h of the treatment. Plants were further grown for 3–5 weeks until the siliques were brown and dry. Seeds were collected after gentle dragging of the inflorescence with finger over the piece of clean paper. The pod and stem material were removed after gentle blowing. Seeds were kept at 4 °C in microcentrifuge tube under desiccation.

Transgenic plants (T_1_ generation) grown on the soil were selected by spraying with 1% BASTA herbicide after two weeks of germination. This procedure was repeated four times after every two days interval. Transgenic plants were easily identified at the end of the BASTA selection. Transformed plants continued to grow and remained green. The untransformed plants remained small, became white, and died two weeks after selection. Seeds were harvested from the treated plants after the maturation of the siliques. Upon acquiring the T_2_ plants of transgenic Arabidopsis, homozygous plants were selected by growing multiple lines and germinating seeds from all lines after 1% basta (glufosinate ammonium) selection. The T_3_ generation was obtained from homozygous lines of three synthetic promoter modules (SynP15, SynP16 and SynP18). The T_3_ transgenic plants were used further for further analysis.

### 4.7. Analysis of Transgenic Arabidopsis

Genomic DNA was extracted from the 3 weeks old T_3_ transgenic Arabidopsis plants using CTAB method. One gram of leaf tissue was ground to a fine powder in liquid nitrogen with mortar and pestle. The fine powder was transferred to 2 mL of pre-warmed CTAB buffer in a 5 mL centrifuge tube. The samples were vortexed and incubated at 65 °C for 1 h with occasional mixing. Equal volume of isoamyl alcohol: chloroform mixture (24:1) was added and mixed gently by inverting the tube several times. The samples were centrifuged at 12,000 rpm for 15 min at 4 °C and the aqueous phase was carefully removed into clean centrifuge tube. A 0.7 volume of isopropanol was added in extracted DNA and precipitated at room temperature, centrifuged at 12,000 rpm for 10 min at 4 °C. The supernatant was removed. The pellet was air dried and resuspended in 50 µL of TE.

A 10 µL of RNase A from stock of 10 mg/mL (Fermentas, Saskatchewen, Canada) was added to DNA solution. The mixture was incubated at 37 ℃ for 30 min. Equal volume of chloroform and phenol (1:1) was added to it and centrifuged at 13,000 rpm for 10 min. The upper aqueous phase was transferred to a fresh centrifuge tube. Equal volume of chloroform was added and centrifuged at 4 °C. The aqueous phase was removed and this step was repeated again. The DNA was precipitated by the addition of 1/10 volume of 3 M sodium acetate (pH 5.2) and 2.5 volumes of chilled absolute ethanol. The DNA was pelleted at 15,000 rpm for 15 min by centrifugation, washed with 70% ethanol, the pellet was air dried and suspended in 50 µL of TE buffer for further experiments. Quantification of DNA was carried out by agarose gel analysis and spectrophotometric measurement.

PCR amplification of synthetic promoter sequences with GUS specific primers was carried out. The reaction was performed using 25 ng of template DNA, 0.25 μM each of the primers and 1 unit of Taq Polymerase enzyme (TAKARA) along with water. The PCR product was separated on 1% agarose gel. All the putative transgenic plants were screened by PCR using promoter::GUS fusion primers.

### 4.8. Gus Assays of Drought-Treated Transgenic Arabidopsis

To characterize the inducibility of synthetic promoters in response to drought stress, 20-day-old transgenic Arabidopsis plants were used for the following treatment. The roots of plants were immersed in 20% PEG6000 solution for 24 h. For drought inducible expression in other parts, mature plants were subjected to drought stress by withholding water for ten days and GUS expression was checked in flowers and siliques for drought inducible expression. Histochemical assay was performed for three promoter constructs (SynP15, SynP16, and SynP18) to check GUS expression. The treated transgenic plants were also frozen in liquid nitrogen and stored at −80 °C for protein extraction and GUS fluorometric assays. The control plants were incubated in water and wild type Arabidopsis plants served as a negative control. Transgenic plants were analyzed for drought inducible tissue-specific GUS expression through histochemical and fluorometric assays.

## 5. Conclusions

After in silico analysis of 1000 bp promoter sequences of 63 drought-responsive genes, the putative drought-specific *cis* elements were selected to design synthetic promoters. The synthetic modules (SynP14, SynP15, Sy16, SynP17, SynP18, and SynP19) were fused with the minimal 35S promoter, a root motif and synthetic intron and cloned in pCAMBIA3301 by replacing CaMV 35S promoter to drive GUS gene. The results showed that the promoters were active in roots when plants confronted drought stress. In transgenic *Arabidopsis,* SynP16 exhibited the highest root-specific expression against drought among all the synthetic promoters. The critical parameters for designing synthetic promoters are *cis* motifs selection, their copy number, orientation, and spacing. We also confirmed that the *cis* acting sequence termed as the root specific motif was active for tissue-specific expression in synthetic modules. The present study has resulted in the construction and functional characterization of a novel combination of *cis* motif sequence that can drive significantly higher expression in transgenic hairy roots in soybean and *Arabidopsis* as compared to commonly used promoters. However, it is recommended that specific mutations within *cis* acting motifs core would better demonstrate the binding transcription factors to boxes.

## Figures and Tables

**Figure 1 ijms-21-01357-f001:**
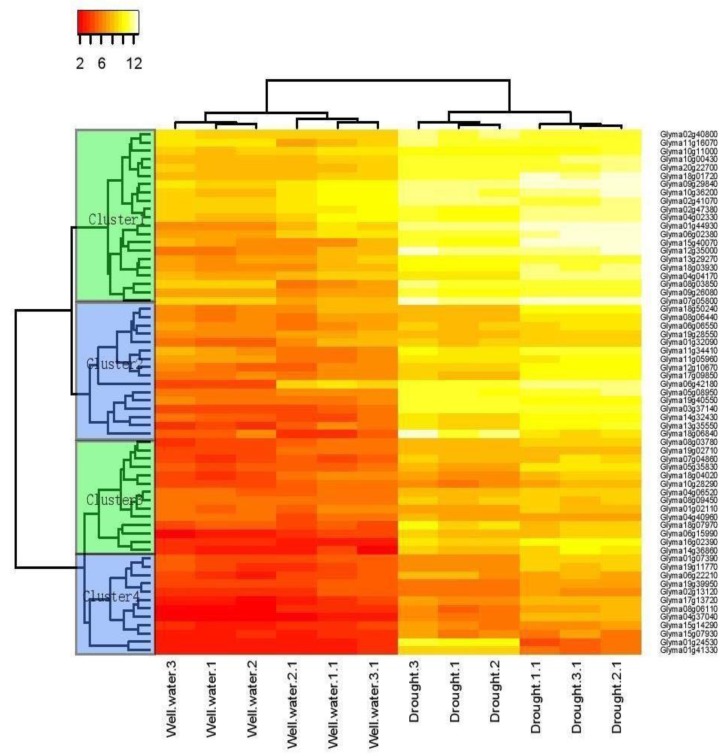
Heat map showing genes upregulated during drought. A total of 63 genes (Appendix A) upregulated during drought stress and four clusters were obtained by using expression spectrum. In order to obtain optimal drought-responsive genes, the intersection of genes upregulated in both studies (ZT0, Z4 [41]; and R2, V6 [42]) was finally used as the final candidate drought corresponding genes. By processing the chip data in the literature [42] to obtain the drought-responsive gene expression profiles under different conditions, and then clustering the expression profiles to obtain four clusters.

**Figure 2 ijms-21-01357-f002:**
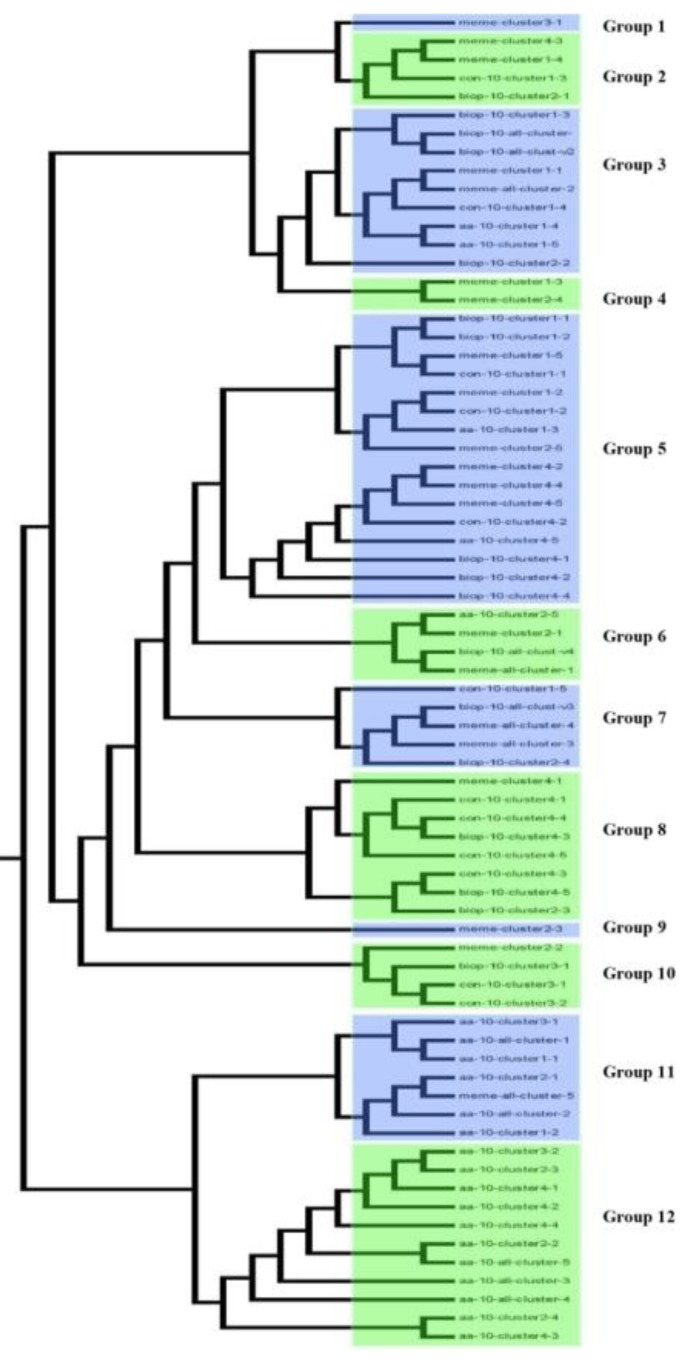
Classification of motifs using MEGA. All 73 motifs clustered into 12 groups.

**Figure 3 ijms-21-01357-f003:**
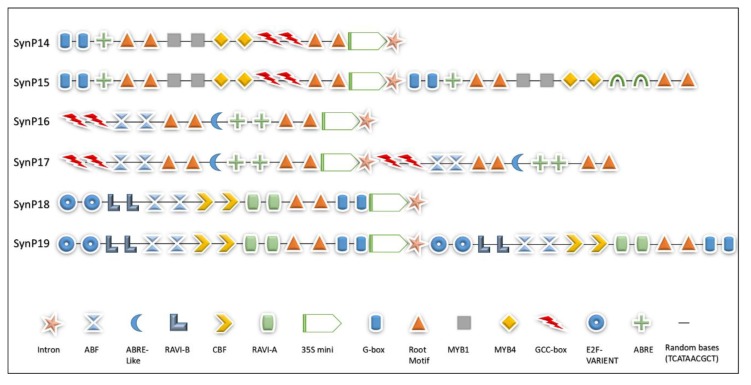
Empirical design of synthetic assemblies. All six promoters (SynP14–SynP19) comprised of 11 multiple *cis* motifs (colored shapes) spaced by random bases (black line) and a root motif. SynP14 contained multiple drought-inducible *cis* motifs upstream of 35S minimal promoter while SynP15 contained same synthetic module both at the upstream and downstream of 35S minimal promoter region. SynP16 contained another set of drought-inducible *cis* motifs. SynP17 contained same synthetic module both at upstream and downstream of the 35S minimal promoter region. SynP18 is the third set of *cis* motifs active against drought stress. Synp19 contained same synthetic module both at the upstream and downstream of 35S minimal promoter region.

**Figure 4 ijms-21-01357-f004:**
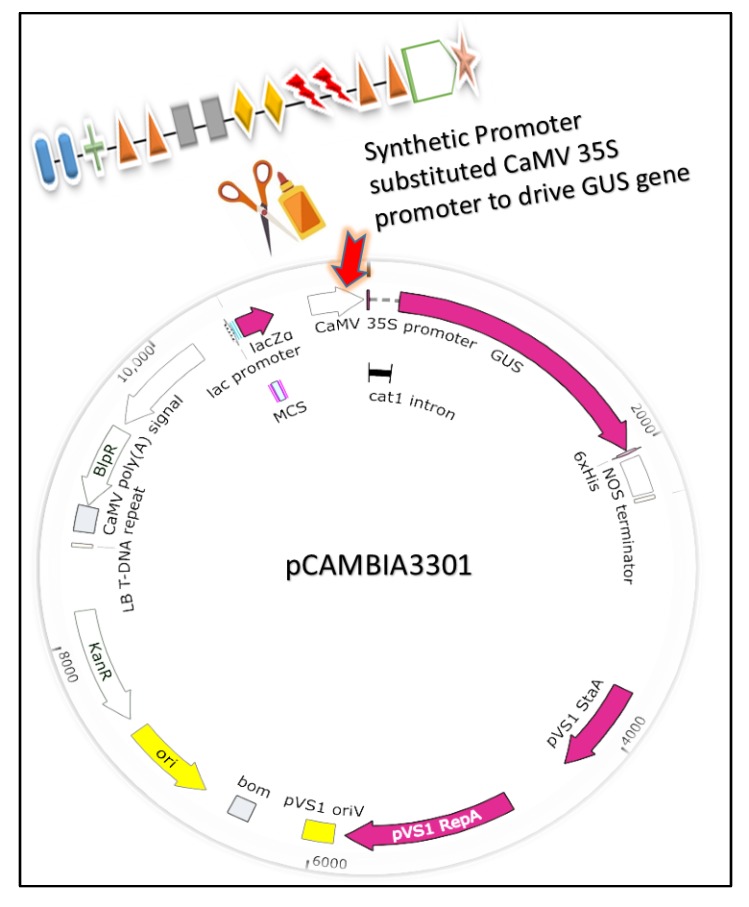
Schematic representation of the construction of expression cassette. Synthetic promoter was inserted by replacing the 35s promoter among the *NcoI* and *HindIII* restriction sites in pCAMBIA3301.

**Figure 5 ijms-21-01357-f005:**
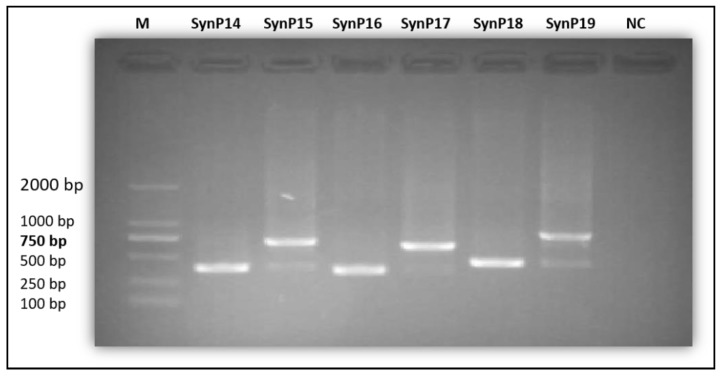
PCR verification of K599 transformants. M is 2 kb Marker and NC is negative control.

**Figure 6 ijms-21-01357-f006:**
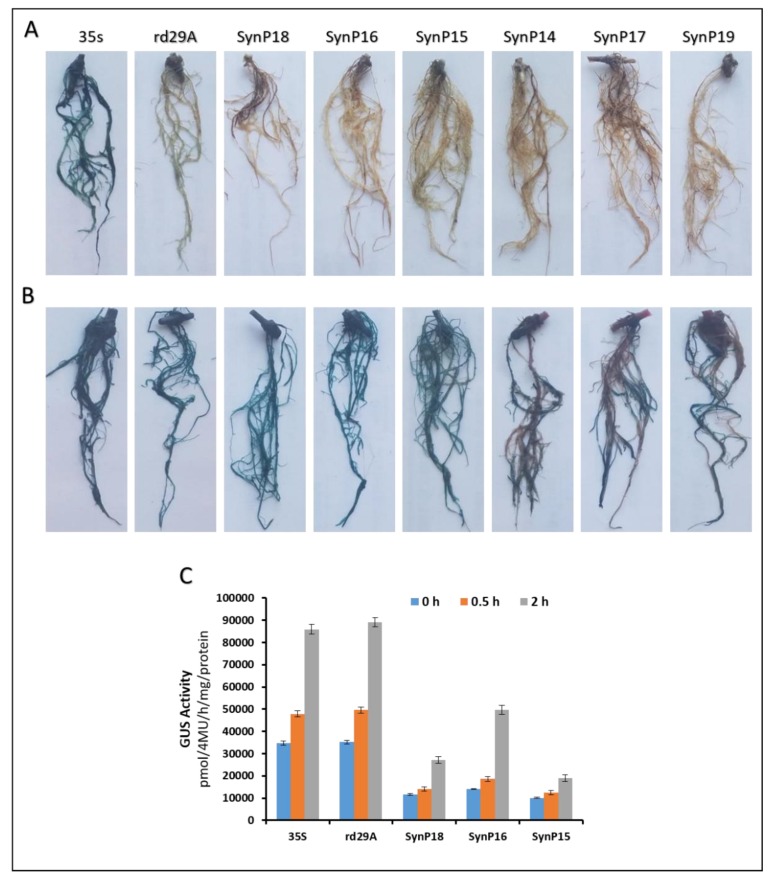
Functional characterization of synthetic promoters in soybean hairy roots. Six synthetic promoters along with control vectors (35S and rd29a as controls) were transformed into soybean via *Agrobacterium rhizogenes*-mediated transformation. Through GUS histochemical staining of ~10 cm long roots (**A** and **B**) and Fluorometric Assay of total protein (**C**), the drought-inducibility of expression cassettes was evaluated. GUS staining before (**A**) and after (**B**) drought treatment. (**C**) GUS fluorometric assay for the 3 efficient synthetic promoters with 35S and rd29a as positive controls after drought treatment. As obvious from figures A and B, GUS staining was not very profound in SynP14, SynP17, and SynP19 thus only SynP15, SynP16, and SynP18 were proceeded to GUS quantification. Hairy roots shown here are of ~30 days old soybean plants.

**Figure 7 ijms-21-01357-f007:**
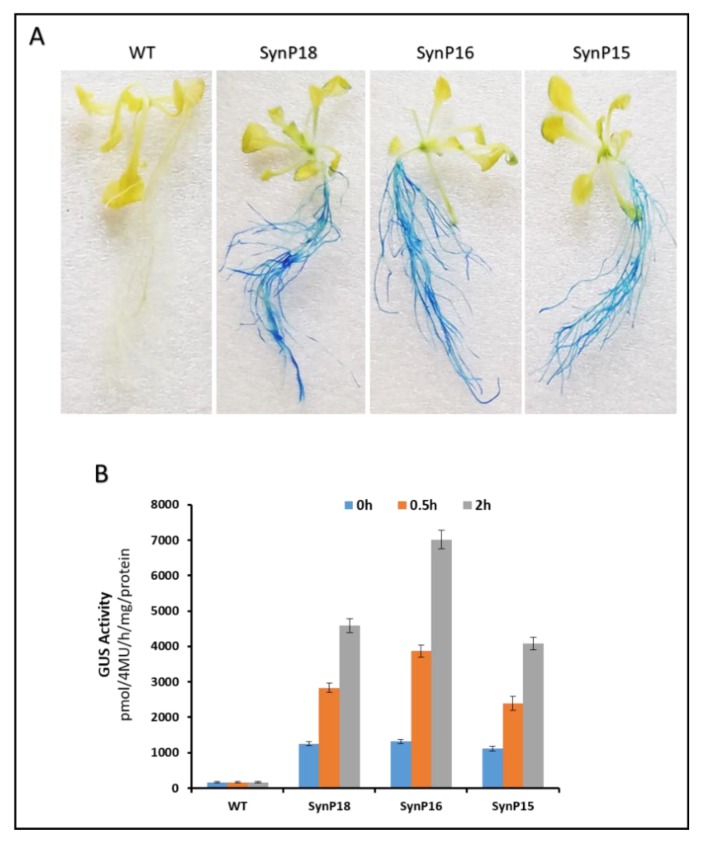
Functional validation of synthetic promoters in Arabidopsis. The three synthetic promoters (SynP15, SynP16, and SynP18), which displayed efficient drought inducibility in soybean hairy roots were further validated in Arabidopsis through their *Agrobacterium tumefacians*-mediated transformation (WT Arabidopsis as negative control). (**A**) WT as well as transgenic Arabidopsis were subjected to GUS histochemical assay after treating with 20% PEG6000 for 24 h. (**B**) GUS fluorometric analysis of WT and transgenic Arabidopsis.

**Table 1 ijms-21-01357-t001:** Elements and their sequences used in construction of synthetic promoters. The intron is of *GmEF-1a* (elongation factor-1a) gene from soybean.

Name	Sequence
***ABF*** [21]	CACGTGGC
***ABRE*** [35]	TACGTGGC
***ABRE-LIKE*** [54]	ACGTGTC
***CBF*** [55]	TGGCCGAC
***E2F-VARIANT*** [56]	TCTCCCGCC
***G-BOX*** [57]	CACGTG
***GCC-BOX*** [58]	GCCGCC
***MYB1*** [59]	TCCTACC
***MYB4*** [60]	ACCTACC
***RAV1-A*** [61]	CAACA
***RAV1-B*** [61]	CACCTG
**RANDOM BASES**	TCATAACGCT
**ROOT MOTIF**	ATATT
**35S MINI**	TCTCCACTGACGTAAGGGATGACGCACAATCCCACTATCCTTCGCAAGACCCTTCCTCTATATAAGGAAGTTCATTTCATTTGGAGAGGA
**INTRON**	CCAGATCTTATATAAGTTTTTGGTTCAAGAAAGTTTTTGGTTACTGATGAATAGATCTATTAACTGTTACTTTAATCGATTCAAGCTAAAGTTTTTTGGTTACTGATGAATAGATCTATTATCTGTTACTTTTAATCGGTCAAGCTCAAGTTTTTTGGTTACTGATGAATAGATCTATATACGTCACAGTGTGCTAAACATGCCCTTGTTTTATCTCGATC

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
