# Peer review of "Tinkering Cis Motifs Jigsaw Puzzle Led to Root-Specific Drought-Inducible Novel Synthetic Promoters"

_ijms, 2020, doi:10.3390/ijms21041357_

Round 1

Reviewer 1 Report

The work presented for review is well written and above all made with very advanced and well-chosen methods. The publication is very specialized and focused on obtaining a synthetic promoter as well as focused on cis motif analysis.

Authors should both better emphasize the achievements of these advanced studies and also describe in the introduction the mechanisms that plants can use in drought conditions. Will the obtained results really find practical application? For which plants and to what extent, in what areas?

At the end of the introduction subchapter there should be a clear hypothesis and the purpose of the work.

The mechanisms concern morphological changes (ramified root system and

increased root-to-shoot ratio) as well as physiological (the accumulation of several hormones and osmotic modulators, such as abscisic acid (ABA), ethylene, hydrogen peroxide and polyamines, the composition of plant pigments: chlorophylls, carotenoids and anthocyanins)

The authors should also mention the difference between the drought defense mechanism in monocotyledons and dicotyledonous. Can the results obtained in this work for a dicotyledonous plant be applied to monocotyledonous plants?

All this should be described in a very synthetic way in the introduction and related to the obtained results described in this paper (the discussion part).

Fig. 5 The individual samples should be clearly marked with symbols, as in Fig. 3

Fig. 6. Parts A, B and C should be consistent - photos should be presented in the order shown in diagram C. If results for SynP14, SynP17, and SynP19 are not shown in this chart, this should be clearly indicated and otherwise explained. The legend should include a description of the seedlings of plants whose roots are presented in the pictures: age, and how they were obtained - growth conditions. In the description in the Materials and Methods subchapter it is difficult to find (lines 442-443?)

Author Response

Authors should both better emphasize the achievements of these advanced studies and also describe in the introduction the mechanisms that plants can use in drought conditions. The mechanisms concern morphological changes (ramified root system and increased root-to-shoot ratio) as well as physiological (the accumulation of several hormones and osmotic modulators, such as abscisic acid (ABA), ethylene, hydrogen peroxide and polyamines, the composition of plant pigments: chlorophylls, carotenoids and anthocyanins)

Thank you for your kind suggestion. We have emphasized the achievements and applications of our studies. We have also modified the Introduction section by describing the mechanisms that plant can use in drought conditions. Details concerning morphological as well as physiological changes in response to drought have been included.

Will the obtained results really find practical application? For which plants and to what extent, in what areas?

Thank you your kind suggestion.

Our designed synthetic promoters are short and could be easily cloned for genetic engineering of plants.

The 6 synthetic promoters reported in manuscript have been tested in triplication. Thus we suppose that these promoters especially SynP16 could be exploited in experiments where a root-specific and drought-inducible expression is desirable.

As the cis motifs we used in designing synthetic promoters are from soybean genes and we tested the expression cassettes in soybean hairy roots and Arabidopsis. Both of the plants are dicots thus we could only say that these synthetic promoters are applicable in genetic engineering of dicots. However, we suggest to test their applicability in monocots.

Our designed promoters have implication in developing drought-tolerant varieties of important crop plants. In contrast to the constitutive expression like that of 35S promoters, we at times need stimulus-inducible and root-specific expression of a gene, thus these promoters are specific to root and are induced only under drought stress. A temporarily regulated gene which is expressed in a specific tissue or even in a specific organelle might produce a better effect than a rampant overproduced protein disturbing the environment or clogging the protein synthesis machinery. Our results are confined to lab-based research and should be tested in the field.

At the end of the introduction subchapter there should be a clear hypothesis and the purpose of the work.

Dear Sir, thank you your kind suggestion. We have revised and added hypothesis and purpose of our work at the end of the introduction subchapter.

The authors should also mention the difference between the drought defense mechanism in monocotyledons and dicotyledonous. Can the results obtained in this work for a dicotyledonous plant be applied to monocotyledonous plants?

Thank you your kind suggestion.

As the cis motifs we used in designing synthetic promoters are from soybean genes and tested the expression cassettes in soybean hairy roots and Arabidopsis. Both of the plants are dicots thus we could only say that these synthetic promoters are applicable in genetic engineering of dicots. However, their applicability is suggested to test in monocots.

All this should be described in a very synthetic way in the introduction and related to the obtained results described in this paper (the discussion part).

Thank you your kind suggestion. Introduction and discussion section have been revised.

Fig. 5 The individual samples should be clearly marked with symbols, as in Fig. 3

Thank you your kind suggestion. We have revised Fig. 5. and the individual samples have been marked clearly with symbols.

Fig. 6. Parts A, B and C should be consistent - photos should be presented in the order shown in diagram C. If results for SynP14, SynP17, and SynP19 are not shown in this chart, this should be clearly indicated and otherwise explained. The legend should include a description of the seedlings of plants whose roots are presented in the pictures: age, and how they were obtained - growth conditions. In the description in the Materials and Methods subchapter it is difficult to find (lines 442-443?)

Thanks for pointing out the shortcomings. We have revised fig. 6 by correcting the order of pictures.

For SynP14, SynP17 and SynP19, we have explained the reason for not including their results in the chart.

We have included the description of soybean seedlings whose roots are presented. They were ~30 days old and methodology of obtaining them is present in Materials and Methods subchapter.

Reviewer 2 Report

In the manuscript of ‘Tinkering Cis Motifs Jigsaw Puzzle Led to Root- Specific Drought-Inducible Novel Synthetic Promoters’, Jameel et al. identified several cis-elements responsible for drought reduction by using transcriptomics-based approach. Moreover, they selected 12 of them for generating 6 synthetic promoters and tested the strength of drought induction both in soybean and Arabidopsis. Here are a few specific comments:

The transcriptomics data is done in this study or derived from previous study? If the authors did the transcriptomic analysis, please add the information including detail of drought treatment in the Materials and Methods section. If it was derived from previous study, please cite the reference. Please make a note about the sample name in figure 1 and illustrate how to get the fold change. The authors clustered 63 drought-inducible genes to 4 groups based on the result of clustering. The motifs of interest were clustered to 12 groups. What are the characters and meaning of these groups? Since they used all the genes for identifying cis-elements and picked 12 motifs for further experiments, why did they need to emphasize the clustering data? Line 125-126: Many cis motifs were not really being associated with abiotic stress response.
Do they mean the cis motifs identified in this study? If so, these motifs are derived from the promoter region of drought-inducible genes. How do they know that these motifs don’t associate with abiotic stress? Line 132-133: Among 73 motifs, 13 were selected which are reported to be involved in drought
(1) Please cite the reference.
(2) Is there any new element identified associated with drought stress? Are the 12 motifs used for generating synthetic promoters selected from the 13 well-studied ones? If so, why just selects 12 instead of 13? How to select the motifs for generating each synthetic promoter? How to determine the copy number and the order of motifs in the synthetic promoter? The information about the intron inserted in the synthetic promoter should be mentioned in the Result section. The figure 6 showed that the promoter of rh6A was highly induced in all the transgenic roots while the synthetic promoters were partially induced. Does that mean rh6A is good enough for the further study and application in the future? The authors used PEG to mimic drought stress but actually PEG treatment induces osmotic stress. May the authors explain why they use PEG?

Author Response

Respected Sir, Thank you for critically reviewing the manuscript. Your comments have greatly improved the quality of our manuscript. We have revised our manuscript according to your kind suggestions.

The transcriptomics data is done in this study or derived from previous study? If the authors did the transcriptomic analysis, please add the information including detail of drought treatment in the Materials and Methods section. If it was derived from previous study, please cite the reference.

The transcriptomics analyses was derived from previous study, however we analyzed their data, selected 63 genes and carried out all bioinformatics analyses for cis motifs selection.

Please make a note about the sample name in figure 1 and illustrate how to get the fold change. The authors clustered 63 drought-inducible genes to 4 groups based on the result of clustering. The motifs of interest were clustered to 12 groups. What are the characters and meaning of these groups? Since they used all the genes for identifying cis-elements and picked 12 motifs for further experiments, why did they need to emphasize the clustering data?

The necessary details of samples have been included in figure legend. The clustering of genes and motifs was done to check their similarity. The members (gene/motifs) of each group indicate higher mutual similarity. After motif analyses, we picked 11 motifs for our experiment but the rest of the analyzed data is not useless. More motifs from each group of clustering data are also being tested, the results of which yet incomplete.

Line 125-126: Many cis motifs were not really being associated with abiotic stress response.
Do they mean the cis motifs identified in this study? If so, these motifs are derived from the promoter region of drought-inducible genes. How do they know that these motifs don’t associate with abiotic stress?

Appendix A shows all the motifs which were screened out of the 63 selected genes. All these 73 motifs clusters were divided into 12 groups and when analyzed with various software, we noticed that many cis motifs were not associated with abiotic stress response. Although these motifs were present in the drought-responsive genes, however, in each gene, there is not only a single cis motif but many, some of which might be associated with other functions and not only abiotic stress.

Line 132-133: Among 73 motifs, 13 were selected which are reported to be involved in drought 
(1) Please cite the reference.

Among 73 motifs clusters, we selected 11 drought-inducible motifs (G-box, MYB4, ABRE, ABRE-Like, RAVI-B, GCC-Box, RAVI-A, E2F-VARIANT, MYB1, ABF and CBF). (The text has been corrected in the revised manuscript) Majority of these motifs have been reported to be involved in abiotic stress specifically drought. The reference has been cited.

(2) Is there any new element identified associated with drought stress?

No new element associated with drought stress is identified.

Are the 12 motifs used for generating synthetic promoters selected from the 13 well-studied ones? If so, why just selects 12 instead of 13?

Thank you for pointing out this textual mistake. Actually, not 13, but 11 cis motifs were selected for the construction of synthetic promoters. The mistake has been corrected throughout the manuscript.

How to select the motifs for generating each synthetic promoter?

The cis motifs reported to be involved in abiotic stress (drought) in previous literature were selected.

How to determine the copy number and the order of motifs in the synthetic promoter?

The copy number and order of cis motifs in designing synthetic promoters do not follow strict rules but as the title of the manuscript indicates (jigsaw puzzle), it’s a trial and error based approach and could face failures many times provided that the basic structure doesn’t alter, i.e.; core, proximal and distal promoters. This may be considered a combined and modified protocol described in previous literature.

The information about the intron inserted in the synthetic promoter should be mentioned in the Result section.

The intron sequence was taken from elongation factor1a gene from soybean (has been mentioned in the manuscript).

The figure 6 showed that the promoter of rh6A was highly induced in all the transgenic roots while the synthetic promoters were partially induced. Does that mean rh6A is good enough for the further study and application in the future?

The rd29a promoter is already in use for drought-inducible expression (also used as a control) and has proved quite efficient however, it’s not tissue-specific.

The authors used PEG to mimic drought stress but actually PEG treatment induces osmotic stress. May the authors explain why they use PEG?

In previous literature, PEG is being frequently used to create simulated drought/osmotic stress.

Reviewer 3 Report

Aysha Jameel et al, used Six  synthetic promoters  with  cis motifs, ARFs, ABRE, ABRE-Like, CBF, E2F-VARIANT, G-box, GCC-Box, MYB1, MYB4, RAVI-A and RAVI-B and a minimal 35s core promoter and a 222 bp synthetic intron
17 sequence.  The results showed that the promoters were active in roots of  drought stress plants. Moreover, authors showed that cis motifs selectionated,  copy numbers, orientation and spacing are important for its expression ehnanced. The paper is well written and the results are well discussed. Hawever, i thinck that  specific mutations within cis-acting motifs core  would have been desiderable to demonstrated the binding transcriptor factors to boxes.I understand that further experiments need further work. However, the authors could include this aspect in the conclusions as future prospects.

Best regards

Author Response

Dear Sir, thank you for taking your time to review our manuscript. We have included your kind suggestions in the conclusions section.

Reviewer 4 Report

dear authors, manuscript describes interesting results about specific drought-inducible synthetic promoters. Manuscript is written well, however there are some comments that should be considered before this manuscript might be accepted.

Abstract

line 22: change "for a dual verification"

line 24: change interestingly

Introduction

lines 35-37: it's not clear

lines 48-52: improve references

lines 99-102: change/move on to the conclusions

Results: figure 6-7. add bar to the scan of roots. about barplot you didn't provide statistic. figure 7, wt values are not clear, change barplot

Discussion

line 254-257. add references

Materials and methods

lines 360-362: change and improve references

line 395: E.coli change in  Escherichia coli

line 488: Arabidopsis (italic)

line 491: check jxem2sec

Author Response

Dear teacher, thank you for critically reviewing the manuscript. We have made the modifications in our manuscript according to your kind suggestions.

Abstract

line 22: change "for a dual verification"

Ok Sir. Thank you. We changed to “Afterwards”

line 24: change interestingly

Ok Sir. We have changed to “Consistently”

Introduction

lines 35-37: it's not clear

Ok Sir. Thank you. The sentence has been clarified.

lines 48-52: improve references

Ok Sir. Thank you. We have included the reference.

lines 99-102: change/move on to the conclusions

Ok Sir. Thank you. We have moved to conclusions.

Results:

figure 6-7. add bar to the scan of roots. about barplot you didn't provide statistic.

Ok Sir. Thank you. We have mentioned the length of soybean hairy roots at the time of GUS staining in the legend of figure 6.

figure 7, wt values are not clear, change barplot

Ok Sir. Thank you. The wt values were negligible thus aren’t visible in figure 7.

Discussion

line 254-257. add references

Ok Sir. Thank you. We have added the references.

Materials and methods

lines 360-362: change and improve references

Ok Sir. Thank you. We have revised and included the references.

line 395: E.coli change in  Escherichia coli

Ok Sir. Thank you. We have changed to Escherichia coli.

line 488: Arabidopsis (italic)

Ok Sir. Thank you. We have italicized Arabidopsis.

line 491: check jxem2sec

Ok Sir. Thank you. We have corrected.